# Research on Pavement Performance of Cement-Stabilized Municipal Solid Waste Incineration Bottom Ash Base

**DOI:** 10.3390/ma15238614

**Published:** 2022-12-02

**Authors:** Chenglin Shi, Jia Li, Shuang Sun, Hong Han

**Affiliations:** 1School of Transportation Science and Engineering, Jilin Jianzhu University, Changchun 130119, China; 2Changchun Highway Administration, Changchun 130021, China

**Keywords:** pavement base, municipal solid waste incineration bottom ash, mechanical properties, frost resistance, constitutive model

## Abstract

In order to clarify the influence of the municipal solid waste incineration bottom ash (MSWI BA) content on the pavement performance of the cement-stabilized macadam, the MSWI BA with 0%, 25%, 38% and 50% content was used instead of fine aggregates. To explore the feasibility of building pavement base with cement stabilized MSWI BA, the cement-stabilized MSWI BA mixture was prepared by mixing the MSWI BA at the mass fraction of 50%, 75% and 100% with fine crushed stuff. Subsequently, the compaction test and 7 days unconfined compression test were conducted with 4%, 5% and 6% cement dosage. The compaction test, unconfined compressive strength test, splitting strength test, compressive resilient modulus test and frost resistance tests were carried out based on the long-age samples with an optimal cement dosage of 5%. Furthermore, the unconfined compressive constitutive model was established based on the test data. Afterwards, the test road was built to measure the practical effect of MSWI BA on road construction. Meanwhile, energy-saving and emission-reduction analyses were conducted on the MSWI BA road. The results showed that under 5% cement dosage, the mechanical properties and frost resistance of the mixture with different MSWI BA content both satisfied the specification requirements; during the construction, the appropriate MSWI BA content could be selected according to the requirements of different highway grades in the specification. The established segmented constitutive model could well simulate the stress–strain relationship of the mixture in the compressive process. Using cement-stabilized MSWI BA to build the pavement base was feasible, which provided not only an important reference for the engineering design but also had positive significance for promoting carbon peaking, carbon neutrality and sustainable development of highway engineering construction.

## 1. Introduction

The Chinese government has put forward the goal of peaking carbon dioxide emissions by 2030 and achieving carbon neutrality by 2060. In this context, the Development Planning of Municipal Solid Waste Classification and Treatment Facilities in the 14th Five-Year Plan was issued by the Chinese government, which proposes that by the end of 2025, the incineration capacity of municipal solid waste in China will reach the target of approximately 8 × 10^6^ t/d. Municipal solid waste incineration power generation can make full use of the calorific value of waste and eliminate a large number of harmful bacteria and toxic substances it contains [1]. After incineration, waste can be reduced in weight and volume by 70% and 90%, respectively [2]. Therefore, incineration is an ideal waste treatment method. However, substantial residue is left over from waste incineration, with municipal solid waste incineration bottom ash (MSWI BA) accounting for about 80% of the total [3], which not only occupy land resources but also have adverse effects on the natural environment if piled up or landfilled at will [4].

MSWI BA is classified as general solid waste in China and has little impact on the environment, which provides a certain technical guarantee for its resource utilization [5,6]. MSWI BA contains cement clinker minerals, active SiO_2_ and Al_2_O_3_ [7], which have certain gelling activity, can take place in hydration reaction and pozzolanic reaction [8], as well as contributes to the improvement of the cohesion of materials [9]. Their mechanical properties are similar to dense sand [10]. MSWI BA is an excellent road material, which can be used to treat poor soil [11], fill subgrade and act as aggregate in asphalt mixture, cement concrete and pavement base [12,13,14]. Scholars have conducted a lot of research on the application of MSWI BA in cement-stabilized macadam bases [15,16,17]. MSWI BA is usually used to replace a part of the natural aggregate. The incorporation of MSWI BA will adversely affect the mechanical properties and frost resistance of the mixture but can improve the dry shrinkage performance [18].

In China, many areas are facing the problem of a shortage of sand. Therefore, lime and cement-stabilized fine-grained soil are often used as the pavement base in the construction of low-grade highways [19], which is relatively expensive and not durable; thus, it seriously affects the service life of the road. Building the pavement base without using natural aggregates, only using cement-stabilized MSWI BA, will not only improve the performance of the pavement but also solve the problem of MSWI BA disposal.

Due to the different compositions of waste raw materials, incinerator types and incineration methods in different regions, the composition of MSWI BA is different [20]. The MSWI BA from the municipal solid waste incineration power plant in Jilin Province is selected as the research object, used for preparing cement-stabilized MSWI BA macadam mixture. The aim of this paper is to analyze the influence of MSWI BA content on the performance of cement-stabilized macadam base, explore the feasibility of using cement-stabilized MSWI BA to build pavement base and study the mechanical properties and frost resistance of the mixture. Meanwhile, we establish the constitutive model of the compressive process for the mixture. We built a test road to verify the feasibility of a cement-stabilized MSWI BA pavement base. The above research will provide a reliable reference for the design and construction of MSWI BA in pavement base.

## 2. Materials and Methods

### 2.1. Materials

#### 2.1.1. Cement

The cement used is P.C 32.5, produced by Jilin Yatai Cement Co., Ltd. (Changchun, China).

Table 1 shows that the properties of cement meet the requirements of the Common Portland Cement of China (GB 175-2007) [21] and the Chinese specification of JTG F20-2015, Technical Guidelines for Construction of Highway Road Bases [22]. 

#### 2.1.2. Natural Aggregates

The natural aggregate used in the experiment is limestone material, which is divided into 1#, 2#, 3# and 4# on the basis of particle size composition. The gradation composition of the aggregate is shown in Table 2. The basic properties of limestone aggregate are studied based on the Test Methods of Aggregate for Highway Engineering of China (JTG E42-2005) [23], as Table 3 shows. The basic properties of limestone aggregate satisfy the requirements of the Chinese specification of JTG F20-2015.

#### 2.1.3. MSWI BA

The MSWI BA is taken from the municipal solid waste incineration power plant in Jilin Province of China, which is gray-brown (Figure 1) and mainly composed of incineration bottom ash, brick, glass fragments and ceramic fragments. The municipal solid waste is obtained by a series of processes such as incineration, crushing and screening, which have certain particle size distribution. According to the test method of the Chinese specification of JTG 3430-2020, Test Methods of Soils for Highway Engineering [24], the particle analysis of the MSWI BA is conducted. Compared the gradation of the MSWI BA with the coarse sand grading range in the Technical Guidelines for Construction of Highway Cement Concrete Pavements of China (JTG/T F30-2014) [25], the results are shown in Figure 2.

The coefficient of uniformity (*C_u_*) and the coefficient of curvature (*C_c_*) are two necessary parameters to evaluate the aggregate gradation, which are calculated by Equations (1) and (2): (1)$$C_{u}=\frac{d_{60}}{d_{10}}$$
(2)$$C_{c}=\frac{{\left(d_{30}\right)}^{2}}{d_{10}\times d_{60}}$$
where *d*_10_, *d*_30_ and *d*_60_ are characteristic particle sizes (mm) of aggregates. That is, on the particle size distribution curve of aggregate, the mass of aggregate smaller than that particle size is 10%, 30% and 60% of the total mass of aggregate, respectively.

The fineness modulus of MSWI BA is 3.7, which is classified as coarse sand according to the classification of sand. The *C_u_* of MSWI BA is 20.19, and the *C_c_* is 1.05, manifesting that MSWI BA is similar to well-graded sand. The basic properties of MSWI BA are shown in Table 4. The basic properties of MSWI BA meet the requirements of the Chinese specification of JTG F20-2015.

The chemical composition of MSWI BA was analyzed by X-ray fluorescence spectroscopy (XRF). Table 5 shows that the main chemical composition of the MSWI BA is SiO_2_, Al2O_3_ and CaO. Usually, when the sum of the content of SiO_2_, Al_2_O_3_ and CaO in the chemical composition of the material is higher than a certain degree, it will show strong gelling activity. The specification of ASTM C618-2015, Natural Volcanic Ash Raw Materials or Calcined Materials for Fly Ash and Concrete, proposed by the American Association of Materials and Tests, defines the content standard as 70%. The sum of the content of SiO_2_, Al_2_O_3_ and CaO in MSWI BA is much higher than this standard, indicating that MSWI BA has strong gelling activity.

### 2.2. Mixture Design

#### 2.2.1. Cement Stabilized MSWI BA Macadam

Incorporating the MSWI BA into the cement-stabilized macadam to prepare the cement-stabilized MSWI BA macadam base mixture. In order to ensure the stability of the mixture gradation, 4-grade crushed stone and MSWI BA are prepared, as Table 6 shows. The target gradation of the mixture is the medium of the C-C-1 gradation range applicable to the second- and below-order highways in the Chinese specification of JTG F20-2015. Based on the principle that the gradation of the mixture is as close as possible to the target gradation, the maximum MSWI BA content of cement-stabilized macadam is determined by calculation to be 50%. MSWI BA is used to replace 0%, 25% and 38% of the total mass of the mixture, to analyze the influence of MSWI BA content on the performance of cement-stabilized macadam. The mixtures are named CBM-0, CBM-25, CBM-38 and CBM-50, respectively. The mixture gradation is shown in Figure 3.

#### 2.2.2. Cement Stabilized MSWI BA 

The gradation of MSWI BA is excellent, which meets the C-A-4 gradation applicable to the second- and below-order highways Sub-base in Chinese specification of JTG F20-2015. In order to explore the feasibility of cement-stabilized MSWI BA to build a pavement base, incorporating the MSWI BA (50%, 75%, 100%) into the crushed stone fine stuff, named CSB-50, CSB-75 and CSB-100, respectively, as Table 6 shows.

### 2.3. Samples Preparation

The samples are molded by static pressure method based on the Test Methods of Materials Stabilized with Inorganic Binders for Highway Engineering of China (JTG E51-2009) [26]. The cylinder sample of cement-stabilized MSWI BA macadam is Φ 150 mm × 150 mm, and that of cement-stabilized MSWI BA base is Φ 100 mm × 100 mm, then wrapped with plastic film and placed into the standard curing room (Figure 4). The standard curing temperature is 20 ± 2 °C, and the standard curing humidity is 95%. The above molding and curing methods are both suitable for samples used in the unconfined compressive strength test, splitting strength test, compressive resilient modulus test and frost resistance test.

### 2.4. Mechanical Properties Test

#### 2.4.1. Compaction Test

The cement dosages of 4%, 5% and 6% are used for the compaction test. According to the Chinese specification of JTG E51-2009, the optimum moisture content and maximum dry density of cement stabilized MSWI BA macadam are determined by Category C method of heavy compaction test. We use Category A compaction method to determine that of cement stabilized MSWI BA base. 

#### 2.4.2. Unconfined Compressive Strength Test

The 7 days of unconfined compressive strength is an important indicator for the construction quality control and inspection of the pavement base material in Chinese specification of JTG F20-2015. To find the appropriate cement dosage and MSWI BA content, according to the Chinese specification of JTG E51-2009, we conducted 7 days of unconfined compressive strength tests with 4%, 5% and 6% cement dosage, respectively. The test instrument is the Fengyi RFP-03 Intelligent Dynamometer, as Figure 5 shows. The loading rate is 1 mm/min. The maximum pressure is recorded when the sample is damaged. The unconfined compressive strength can be calculated by Equation (3):(3)$$R_{c}=\frac{P}{A}$$
where *R*_c_ is unconfined compressive strength (MPa), *P* is the maximum pressure when the sample is damaged (N) and *A* is the cross-sectional area of the sample (mm^2^).

#### 2.4.3. Splitting Strength Test

The splitting strength is an important indicator for checking the tensile stress at the bottom of the pavement structure, which reflects the failure mode of the material being pulled out. The curing time of the sample is 90 days and the cement dosage is 5%. According to the Chinese specification of JTG E51-2009, the size of the cement-stabilized MSWI BA macadam sample is Φ 150 mm × 150 mm, the corresponding bar width is 18.75 mm, and the arc radius is 75 mm. The size of the cement-stabilized MSWI BA sample is Φ 100 mm × 100 mm, the corresponding bar width is 12.7 mm, and the arc radius is 50 mm. The test instrument is the Lihuan pavement material strength tester LD-127, as Figure 6 shows. The sample is pressurized at the rate of 1 mm/min. The maximum pressure of the sample is recorded when it is damaged. The splitting strength can be calculated by Equation (4): (4)$$R_{i}=\frac{2P}{\pi dh}\left(\sin2\alpha -\frac{a}{d}\right)$$
where R_i_ is the splitting strength (MPa), P is the maximum pressure when the sample is damaged (N), d is the diameter of the sample (mm), h is the height of the sample (mm), a is the width of the bar (mm) and α is the center angle corresponding to the width of the semi-bar (°). 

#### 2.4.4. Compressive Resilient Modulus Test

According to the current design method of highway asphalt pavement in China, the resilient modulus of the semi-rigid base material is an important parameter of the whole pavement design, which directly affects the calculation results of tensile stress and deflection of pavement surface and base. The samples with a curing time of 90 days and cement dosage of 5% are used to conduct the compressive resilient modulus test based on the Chinese specification of JTG E51-2009. The test instrument is the Lihuan pavement material strength tester LD-127, as Figure 6 shows. The loading rate is 1 mm/min. The predetermined unit pressure is divided into six equal parts, increasing the load by step. The resilient deformation of the sample under each load is recorded. The compressive resilient modulus is calculated by Equation (5): (5)$$E_{c}=\frac{Ph}{l}$$
where *E_c_* is compressive resilient modulus (MPa), *P* is unit pressure (MPa), *h* is the sample height (mm) and *l* is the resilient deformation of the sample (mm).

#### 2.4.5. Frost Resistance Test

The winter scene freeze–thaw (F-T) cycles equivalent to indoor 10~14 cycles in Jilin Province [27]; in order to analyze the damage of F-T cycles on pavement base materials, we combined the test methods and evaluation indicators in the specification and the compressive strength ratio of the residual compressive strength after 10 cycles of 180 days sample is selected to characterize the frost resistance.

According to the Chinese specification of JTG E51-2009, the samples are frozen in a low-temperature refrigerator at −18 °C for 16 h, and then melted in a water tank at 20 °C for 8 h, representing a complete F-T cycle. The frost resistance is represented by the compressive strength loss (*BDR*) of the sample after *n* F-T cycles. The calculation method of *BDR* is as Equation (6) shows:(6)$$BDR=\frac{R_{\mathrm{DC}}}{R_{C}}\times 100$$
where *BDR* is the compressive strength loss of the sample after *n* F-T cycles (%), *R*_DC_ is the compressive strength after *n* F-T cycles (MPa) and *R*_C_ is the compressive strength without F-T cycles (MPa).

## 3. Results and Discussion

### 3.1. The Optimum Moisture Content and the Maximum Dry Density

The compaction test results are shown in Figure 7.

Figure 7 shows that with the increase in MSWI BA content, the optimum moisture content of the two-group mixture both increases and the maximum dry density both decreases. The reason is that the density of MSWI BA is lower than that of natural aggregate, and the water absorption rate is higher than that of natural aggregate. Therefore, with the increase in MSWI BA content, the optimum moisture content of the mixture increases, and the maximum dry density decreases. The CSB group is composed of crushed stone fine stuff and MSWI BA. The water absorption of crushed stone fine stuff is larger, and the density is smaller than that of large particle size crushed stone. Therefore, the optimum moisture content of CSB-50 is larger, and the maximum dry density is smaller than that of CBM-50. Under the same MSWI BA content, the changes in optimum moisture content and dry density of the CSB group are also larger.

With the increase in the cement dosage, the optimum moisture content and the maximum dry density of the mixture both increase, but the variations are minimal. Because the amount of cement slurry in the mixture increases with the increasing cement dosage, the increase in cement slurry can not only increase the lubricity between aggregates but also fill in the voids, resulting in the improvement of density. Cement has strong water absorption and cement dosage increases, manifesting the equivalent of increasing the water absorption of the mixture; however, the proportion of cement in the mixture is small, so the effect on the optimum moisture content and the maximum dry density is not significant.

### 3.2. Unconfined Compressive Strength

The 7 days of unconfined compressive strength of mixture with 4%, 5% and 6% cement dosage are shown in Figure 8. 

Figure 8 shows that under the same MSWI BA content, the unconfined compressive strength of the two-group mixture both increases with the increase in cement dosage. The reason is that with the increasing cement dosage, the cement slurry content increases, which in turn, increases the internal bonding force of the mixture, leading to an increase in compressive strength. With the increase in MSWI BA content, the unconfined compressive strength decreases. MSWI BA contains a lot of ceramic fragments and glass fragments, which will break when the sample is molded, forming a weak area inside the sample. At the same time, the surface of ceramic fragments and glass fragments is relatively smooth, and the adhesion between them is poor, which reduces the internal bonding force of the mixture; therefore, the incorporation of MSWI BA has an adverse effect on the unconfined compressive strength. 

The compressive strength of CSB-50 is smaller than that of CBM-50. Under the same changes in MSWI BA content condition, the compressive strength of the CSB group is larger than that of the CBM group. Because of that, in the CBM group, MSWI BA, instead of fine aggregates, fills the voids. The compressive strength in the CBM group is mainly determined by the coarse aggregates, which are embedded with each other to form the skeleton, while that of the CSB group is mainly determined by the internal cohesion of the material. 

The Chinese specification of JTG/T F20-2015 stipulates that the 7 days of unconfined compressive strength of cement-stabilized materials should meet the requirements of Table 7. From above, we know that with the decrease in cement dosage and the increase in MSWI BA content, the 7 days of unconfined compressive strength decreased. Table 7 shows that with 4% cement dosage, the 7 days unconfined compressive strength of the CBM-50 mixture satisfies the requirements of expressways and first-grade highways under medium and light traffic. The CSB-100 can meet the requirements of second- and below-order highways under medium and light traffic. The 7 days of unconfined compressive strength is the main indicator for the construction quality control and inspection of cement-stabilized materials. The results of the 7 days unconfined compressive strength test indicate that it is feasible to build the pavement base with cement-stabilized MSWI BA. The incorporation of MSWI BA makes the compressive strength of cement-stabilized crushed stone decrease. The cement dosage and MSWI BA content can be adjusted as needed in practical applications.

The 7 days unconfined compressive strength of the two groups both increase with cement dosage, while too high a cement dosage will affect the crack resistance of the mixture. In the application of cement-stabilized materials in practical engineering, the commonly used cement dosage is 5%; therefore, a 5% cement dosage is selected to prepare long-age samples for subsequent tests. The unconfined compressive strength at curing times of 28 days, 90 days and 180 days are shown in Figure 9.

Figure 9 shows that the compressive strength increases rapidly from 7 days to 28 days, and the increase rate gradually slows down after 28 days. The increase in the compressive strength is mainly determined by the hydration reaction of cement. The hydration reaction in the mixture at the early curing stage is intense, and then the cement has basically reacted completely at the late curing stage; however, after the incorporation of MSWI BA, the compressive strength still increases greatly at the late curing stage. The 180 days compressive strength of the mixture with 0%, 25%, 38% and 50% MSWI BA in the CBM group increased by 62%, 71%, 76% and 81%, respectively, compared with that of 7 days. The 180 days compressive strength with 100%, 75% and 50% MSWI BA in the CSB group increased by 114%, 91% and 85%, respectively, compared with that of at 7 days. The greater the MSWI BA content, the larger the growth rate. The reason is that the main chemical components of the MSWI BA are SiO_2_, Al_2_O_3_, CaO and other active substances, as Section 2.1.3. concluded, can cause a pozzolanic reaction and play a positive role in strength growth [28]; however, the pozzolanic reaction is relatively slow, so the increasing effect is significant in the late curing stage. While due to the low strength of the MSWI BA itself, the compressive strength in the late curing stage is still lower than that of the control group (CBM-0, CSB-50), even under the action of the active substances in the MSWI BA.

### 3.3. Splitting Strength 

Figure 10 shows that the MSWI BA content increases and the splitting strength gradually decreases. The splitting strength is not only closely related to cohesion but also associated with the strength of aggregate. The MSWI BA contains more glass fragments and ceramic fragments; its surface is smoother than that of the natural crushed stone aggregate; therefore, the cement slurry cannot wrap it well, which reduces the cohesion of aggregates. In addition, the MSWI BA has many edges and corners, and the aggregates in the mixture are squeezed and embedded with each other. The MSWI BA will bear part of the splitting force during splitting, but the strength of MSWI BA is low, which further reduces the splitting strength. The splitting strength of CSB-50 is higher than that of CBM-50. The reason is that CSB-50 is composed of MSWI BA and crushed stone fine stuff, and its specific surface area is larger than that of CBM-50; therefore, the contact area between the cement slurry and the aggregate is larger, so the cohesion of the mixture is larger than that of CSB-50. Otherwise, the size of the samples for the splitting test between the CBM group and the CSB group is different, resulting in a sizable effect. In general, the splitting strength of each group is greater than 0.4 MPa, which satisfied the design reference value in the specification.

### 3.4. Compressive Resilient Modulus 

Figure 11 shows that the MSWI BA content increases, and the compressive resilient modulus of the two-group mixture gradually decreases. The compressive resilient modulus reflects the difficulty of material deformation under load, which is used as an indicator to characterize the stiffness of the material. The compressive resilient modulus can be used to distinguish rigid materials from flexible materials. Compared with crushed stone, the MSWI BA has small density, low hardness and smaller stiffness. The incorporation of MSWI BA will reduce the overall rigidity of the mixture to a certain extent, so with the increasing MSWI BA content, the compressive resilient modulus decreases. The compressive resilient modulus is one of the main parameters in the design of semi-rigid base materials. The modulus decreases after incorporating MSWI BA. If you want to achieve the same bearing capacity as the ordinary base, the thickness of the base needs to be increased.

### 3.5. Frost Resistance

Figure 12 implies that the larger the MSWI BA content, the worse the frost resistance of the mixture. The F-T damage of cement-stabilized base materials is because the water inside the material expands and migrates under F-T, causing damage to the interior of the material; the damage will continue to accumulate with the increasing F-T cycles [29]. The greater the water absorption of the material and the more internal pores, the more serious the damage caused by F-T. The MSWI BA has low density, many pores and high-water absorption. After mixing, the water absorption of the mixture increases and the internal pores increase; therefore, the frost resistance gradually deteriorates with the increase in MSWI BA content. The frost resistance of each group meets the requirements of Technical Specifications for Design and Construction of Highway in Seasonal Frozen Soil Region of China (JTG/T D31-06-2017) [30] for medium and heavy freezing zones; however, the CSB-100 is close to the limit of 70% in the specification.

The effective construction time in the seasonally frozen zone is short. Before winter, the pavement base construction cannot be completed, or the asphalt pavement frequently cannot be laid in time. For the pavement base paving during this time, it is necessary to take antifreeze protection measures to prevent frostbite damage affecting the performance and service life of the road.

## 4. Unconfined Compressive Stress–Strain Constitutive Model

The 90 days unconfined compressive strength stress–strain curves of the mixture are shown in Figure 13. 

Figure 13 shows that with the increasing MSWI BA content, the compressive failure stress gradually decreases, and the failure strain gradually increases. In cement concrete and geotechnical materials, the curved part at both ends of the stress–strain curve characterizes the compaction stage and the elastic–plastic change stage, while the linear part in the middle represents the elastic deformation stage [31,32]. After the incorporation of MSWI BA, the pores in the mixture become more; therefore, with the increase in MSWI BA content, the deformation in the compaction stage gradually becomes larger. Figure 13a shows that the stress–strain curve shape with different MSWI BA content in the CBM group is similar. However, the stress–strain curve with different MSWI BA content in the CSB group is quite different, as Figure 13b shows. Comparing Figure 13a with Figure 13b shows that the stress–strain curve of CSB-50 is similar to that of the CBM group. Because of this, the large-sized crushed stone in the CBM group can be squeezed into each other to form a skeleton structure, while the CSB group cannot form a skeleton structure well, except for CSB-50.

### 4.1. Secant Modulus

The secant modulus, also known as secant stiffness, is defined as the slope of the line connecting a point on the stress–strain curve to the origin. In essence, it is the real-time elastic modulus of the material under load. The ratio of real-time stress to failure ultimate stress, that is, the stress level is used as the abscissa. The real-time secant modulus as ordinate. The variation of the secant modulus with stress levels is shown in Figure 14. 

Figure 14 shows that the secant modulus increases rapidly with the increase in stress level, which corresponds to the bending part in the front of the stress–strain curve, that is, the compaction stage. At this stage, the pores and primary cracks are continuously compressed and closed. With the increase in stress level, the growth of secant modulus slows down and a relatively stable growth exists in the medium term, which corresponds to the linear part of the stress–strain curve; that is, the elastic change stage. At this stage, the stress is continuously concentrated, and the recoverable elastic deformation occurs. The secant modulus gradually decreases after reaching the peak, corresponding to the elastic–plastic failure stage of the strain–strain curve. At this stage, the cracks inside the mixture begin to develop, and the growth rate continues to accelerate. When the ultimate stress and ultimate strain are reached, damage occurs.

### 4.2. Damage Variable

Damage variable *D* is often used to describe the response and damage evolution state of materials under load. Generally, the definition method of damage is selected from macroscopic, mesoscopic and microscopic perspectives. The macroscopic perspective is based on the material macroscopic measurable parameters, such as elastic modulus, wave velocity and acoustic emission parameters. The mesoscopic perspective is based on the mesoscopic statistical damage mechanics method. The proportion of the representative volume element of the damaged material is defined as the damage variable, assuming that the failure volume element obeys a certain probability distribution; then, we established the damage evolution model. The microscopic point of view is to define the material microstructure parameters such as the number of micro-defects, length, area and volume [32].

Lemaitre [33] proposed a common method to characterize the damage evolution of materials under load by using elastic modulus as the damage variable. Damage variable is defined as the ratio of elastic modulus when damage occurs:(7)$$D=1-\frac{E_{d}}{E}$$
where *E*_d_ is the elastic modulus at the material damage state, *E* is the elastic modulus of the material without damage. 0 ≤ *D* ≤ 1, *D* = 0 means that the material has not been destroyed, *D* = 1 means that the material has been completely destroyed.

In geotechnical materials, the elastic modulus under the non-destructive state generally takes the initial elastic modulus of the material, which is more conservative for the cement stabilized MSWI BA, and *D* < 0 will occur, which deviates from the definition of the damage variable. According to the actual situation of the cement-stabilized MSWI BA elastic modulus during the compression process, Equation (8) is used to determine the damage variable *D*:(8)$$D=\left\{\begin{array}{ll} 0 & \sigma <\sigma_{p}\\ 1-\frac{E_{d}}{E_{p}} & \sigma \geq \sigma_{p} \end{array}\right.$$
where *E*_p_ is the peak secant modulus, *E*_d_ is the secant modulus of the elastic–plastic failure stage and *σ*_p_ is the stress level at the peak secant modulus, 0 ≤ *σ*_p_ ≤ 1.

Combined with the definition of the damage variable and Figure 14, it can be seen that the CBM group is damaged at about 0.9 stress level, while the damage stress level of CSB group is quite different; CSB-100 is damaged at 0.6 stress level; therefore, the CBM group have more resistant to load than the CSB group. The CSB group is not suitable for the construction of pavement bases for heavy-traffic roads, so as to avoid damage caused by heavy traffic, the use of the CSB group should be avoided.

### 4.3. Constitutive Model

Scholars are more focused on the constitutive relationship of the material under load [34]. The uniaxial compressive stress–strain constitutive model of cement concrete proposed by Guo is segmented form, in which the ascending section is a polynomial function, and the descending section is a rational fractional function [35]. Yan proposed an improved Duncan-Chang constitutive model to simulate the stress–strain of cement-stabilized macadam during an unconfined compressive process [36]. Zhang established the damage constitutive model of cement-stabilized cinder macadam during the unconfined compressive process through regression analysis [37]. 

Through a large number of simulation tests on the existing models, finding that the existing models cannot well simulate the stress–strain process of the unconfined compression test of cement-stabilized MSWI BA. According to the characteristics of the compressive stress–strain curve of the cement-stabilized MSWI BA mixture, a segmented constitutive model is proposed to describe the compressive process of the cement-stabilized MSWI BA mixture. The constitutive model is divided into three stages, namely, the compaction stage, the elastic change stage and the elastic–plastic failure stage. The compaction stage and the elastic change stage are simulated by polynomial function with reference to the Guo model. In the elastic–plastic failure stage, the damage constitutive model is established based on the statistical damage theory.

According to the strain equivalence hypothesis proposed by Lemaitre, the damage constitutive relationship in the elastic–plastic failure stage can be obtained [38]:(9)$$\sigma =E_{p}\varepsilon \left(1-D\right)$$
where *σ* is the nominal stress, *E*_p_ is the peak secant modulus, namely the reference elastic modulus, *ε* is the strain and *D* is the damage variable.

The development of internal damage to the mixture is complex and random under load, which is difficult to be described by a single characteristic variable. The mixture can be discretized into micro-unit; the damage obeys random distribution and has statistical regular. According to the principle of Weibull distribution, the damage probability density function is expressed as follows [39]:(10)$$\varphi \left(F\right)=\frac{h}{F_{0}}{\left(\frac{F}{F_{0}}\right)}^{h-1}\exp\left[-{\left(\frac{F}{F_{0}}\right)}^{h}\right]$$
where *h*, *F*_0_ are Weibull distribution parameters, *F* is the random distribution variable of micro-unit strength in the mixture.

The *D* in the elastic–plastic failure stage can be defined as the ratio of the number of damaged micro-unit to the total number of micro-units in the mixture from the microscopic point of view:(11)$$D=\frac{N_{d}}{N_{t}}$$
where *N*_d_ is the number of damaged micro-unit in the mixture and *N*_t_ is the total number of micro-units. 

According to Equation (10), the number of damaged micro-unit is:(12)$$N_{d}\left(F\right)=N_{t}\int_{0}^{F}\varphi \left(\varepsilon \right)dDF=N_{t}\left\{1-\exp\left[-{\left(\frac{F}{F_{0}}\right)}^{h}\right]\right\}$$

Then, the damage variable of the mixture in the elastic–plastic failure stage is:(13)$$D=1-\exp\left[-{\left(\frac{\varepsilon }{F_{0}}\right)}^{h}\right]$$

The damage constitutive model of the elastic–plastic change stage of the mixture can be obtained by Equation (9):(14)$$\sigma =E_{p}\varepsilon \exp\left[1-{\left(\frac{\varepsilon }{F_{0}}\right)}^{h}\right]$$

According to the characteristics of the compressive stress–strain curve, the damage constitutive model in the elastic–plastic failure stage should satisfy the following boundary conditions:(15)$$\left\{\begin{array}{c} \frac{d\sigma }{d\varepsilon }{|}_{\varepsilon =\varepsilon_{m}}\\ \sigma {|}_{\varepsilon =\varepsilon_{m}} \end{array}\right.\begin{array}{c} =0\\ =\sigma_{m} \end{array},\left\{\begin{array}{c} E{|}_{\varepsilon =\varepsilon_{p}}\\ D{|}_{\varepsilon =\varepsilon_{p}} \end{array}\right.\begin{array}{c} =E_{p}\\ =0 \end{array}$$
where *ε*_m_ is the ultimate compressive failure strain of the mixture, *σ*_m_ is the failure ultimate stress and *ε*_p_ is the strain at the peak secant modulus. 

Substituting the above boundary conditions into Equations (13) and (14) can obtain Equations (16) and (17) [40]:(16)$$h=\frac{1}{\ln\left(\frac{E_{p}\varepsilon_{m}}{\sigma_{m}}\right)}$$
(17)$$F_{0}=\frac{\varepsilon_{m}}{{\left(\frac{1}{h}\right)}^{\frac{1}{h}}}$$

The segmented constitutive model of stress–strain curve can be expressed by Equation (18):(18)$$\sigma =\left\{\begin{array}{cc} a\varepsilon^{2}+b\varepsilon +c & 0\leq \varepsilon \leq \varepsilon_{s}\\ k\varepsilon +d & \varepsilon_{s}\leq \varepsilon \leq \varepsilon_{p}\\ E_{p}\varepsilon \exp\left[-{\left(\frac{\varepsilon }{F_{0}}\right)}^{h}\right] & \varepsilon_{p}\leq \varepsilon \leq \varepsilon_{m} \end{array}\right.$$
where *ε*_s_ is the strain at the elastic change stage, *ε*_p_ is the strain at the peak secant modulus, *ε*_m_ is the ultimate failure strain and *a*, *b*, *c*, *k*, *d*, *F*_0_, *h* are related model parameters.

The stress–strain curves of CSB-75 and CSB-100 have no obvious linear part. Assuming that the constitutive model is only composed of the compaction stage and elastic–plastic change stage. The relevant model parameters of each group of mixtures are calculated as Table 8 shows.

The comparison between the test curve and the model curve of each group is shown in Figure 15. 

Figure 15 shows that the test curve and the model curve are basically coincident, indicating that the established segmented constitutive model can well simulate the stress–strain relationship of the mixture with different MSWI BA content, which can provide a reference for the study of deformation evolution and mechanical behavior analysis of cement-stabilized MSWI BA pavement base under load.

## 5. Test Road Construction

Through laboratory tests, we know that the mechanical properties and frost resistance of cement-stabilized MSWI BA meet the requirements of the specification. Through the construction of the test road, the practical effect of cement-stabilized MSWI BA to build a pavement base was examined.

The 2020 Rural Road Construction Project in Qianguerluos Mongolian Autonomous County of Songyuan City is the basis of the test road; a 250 m test road was built in Wangfuzhan Village. The test road surface was a cement concrete structure. The test road was a rural road with small traffic. Through the previous laboratory test, the MSWI BA stabilized with 5% cement dosage; the indicators can fully satisfy the requirements of rural roads in the specification. Meanwhile, considering the complexity of construction and economic factors, the pavement base of the test road is built with 5% cement-stabilized MSWI BA, so as to better observe and study the actual performance of MSWI BA for the construction of the pavement base.

The base test section of cement-stabilized MSWI BA was constructed by the road mixing method. First, we spread MSWI BA and cement layer by layer on the road. Then we use road mixers, other machines and manual labor, on-the-spot mixing, to form the structure layer. The main construction process of the cement-stabilized MSWI BA pavement base is shown in Figure 16.

Later observations are made one year after the opening of the test road. The drill core sampling method (Figure 17) is used to examine whether the MSWI BA pavement base could remain intact under repeated F-T cycles and vehicle load. 

After one year of use, the pavement surface is smooth, and there is no disease phenomenon. The core sample is intact, and the side wall is smooth. The above phenomenon proves that the road performance of the MSWI BA base is effective, showing that cement-stabilized MSWI BA to build the pavement base is completely feasible and worthy of large-scale promotion and application.

## 6. Energy Saving and Emission Reduction Analysis

Municipal solid waste incineration power generation mainly reduces carbon emissions through the following two ways: one is to use waste calorific value incineration for power generation to reduce carbon emissions from fossil fuel combustion, and the other is to avoid greenhouse gases generated by landfill through incineration [41,42].

Taking a municipal solid waste incineration power plant in Jilin Province as an example, the designed daily waste treatment capacity of the plant is 1000 t, with a 5 MW generator set. In theory, the annual municipal solid waste treatment capacity is 3.7 × 10^5^ t, and annual power generation is 4.4 × 10^4^ kW·h. According to the data in China Electric Power Industry Annual Development Report 2021 released by the Chinese Electricity Council: the standardized coal consumption for power supply of thermal power plants with a capacity of more than 6000 kW in the country will be 304.9 g/kW·h in 2020, and the CO_2_ emission per unit of thermal power generation in China is about 832 g/kW·h. This shows that a plant can save about 1.3 × 10^5^ t standardized coal and reduce CO_2_ emission by about 3.6 × 10^5^ t in one year. One ton of municipal waste in a landfill will generate 100~200 m^3^ of greenhouse gases [43], mainly composed of CH_4_ and CO_2_, of which CH_4_ accounts for about 45~60% of the total gas. Converting CH_4_ to CO_2_ can yield that one-ton landfill municipal waste will produce 0.2~0.4 t of CO_2_, meaning that plants can reduce CO_2_ generated by landfill by about 7.3 × 10^4~^14.6 × 10^4^ t per year.

In a one-kilometer second-order highway with 9-m width, a base thickness of 40 cm is paved with MSWI BA, the consumption of MSWI BA is about 6.8 × 10^3^ t. One ton of municipal waste can be incinerated to produce about 0.3 t MSWI BA [44]. The comprehensive energy consumption of natural stone mining and processing is equivalent to standardized coal consumption of 6.3 kg/t [45]. Manifesting that the municipal waste from incineration to MSWI BA to build one kilometer second-order highway can save a total of 8.5 × 10^3^ t of standardized coal and reduce 3.2 × 10^4^ t of CO_2_ emissions. At the same time, using MSWI BA to build roads can also reduce environmental damage, soil erosion, noise and smoke caused by natural stone mining and processing. To sum up, municipal waste incineration power generation and MSWI BA road construction have high economic and social value, which are specific actions to actively respond to the national emission reduction policy and closely follow the national green and low-carbon development goals. It has positive significance for achieving a carbon peak in 2030, striving to achieve carbon neutrality in 2060, environmental protection and sustainable development of highway construction.

## 7. Conclusions

(1)With the increase in MSWI BA content, the optimum moisture content increases, the maximum dry density decreases, and the mechanical properties and frost resistance are all adversely affected. The pavement base of higher MSWI BA content should do antifreeze measures in winter.(2)With 4% cement dosage, the CBM-50 satisfies the requirements of expressways and first-grade highways under medium and light traffic. The CSB-100 meets the requirements of second- and below-order highways under medium and light traffic. The cement dosage and MSWI BA content could be adjusted according to the needs of practical application.(3)The established segmented constitutive model can effectively simulate the stress–strain relationship of a mixture in a compressive process, which provides a reliable reference for the design and mechanical behavior analysis of cement-stabilized MSWI BA base.(4)From incineration of municipal solid waste to MSWI BA to build a one-kilometer second-class highway base, we can reduce CO_2_ emissions by about 3.2 × 10^4^ t and save standardized coal 8.5 × 10^3^ t, which has positive significance for promoting carbon peaking, carbon neutrality and sustainable development of highway engineering construction.

## Figures and Tables

**Figure 1 materials-15-08614-f001:**
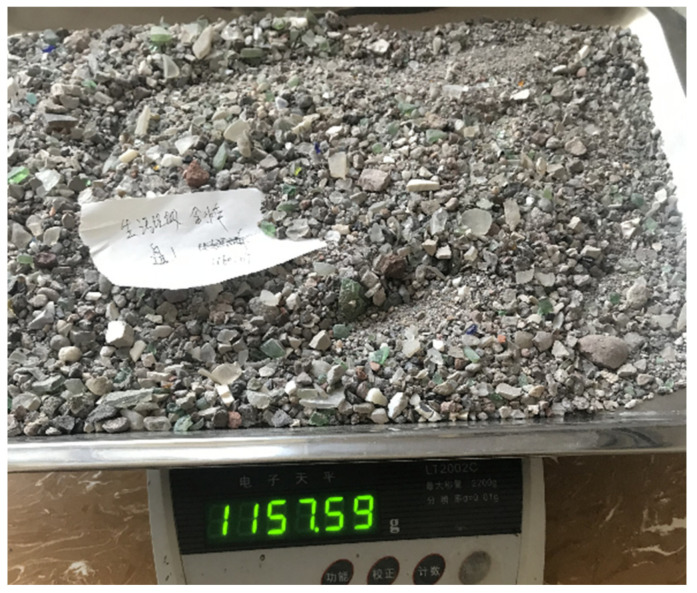
MSWI BA sample.

**Figure 2 materials-15-08614-f002:**
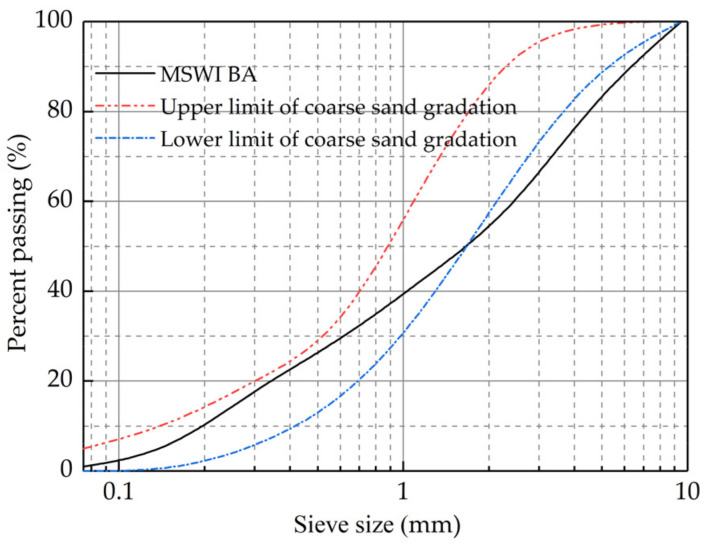
Gradation curve of MSWI BA.

**Figure 3 materials-15-08614-f003:**
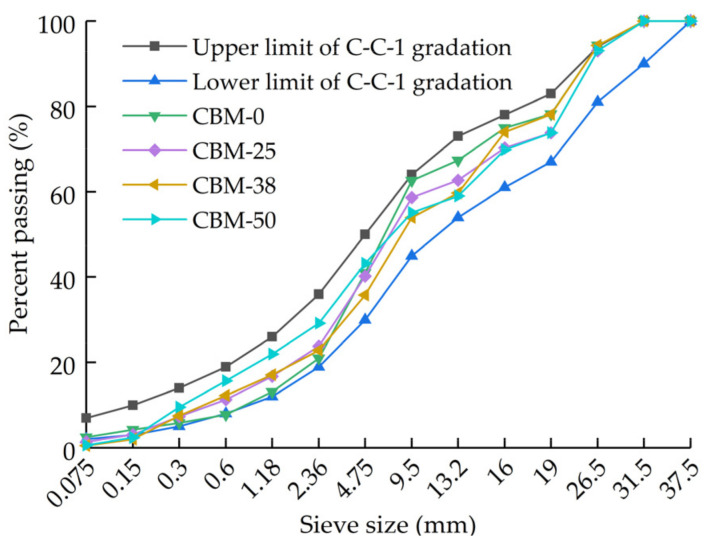
Gradation curve of CBM.

**Figure 4 materials-15-08614-f004:**
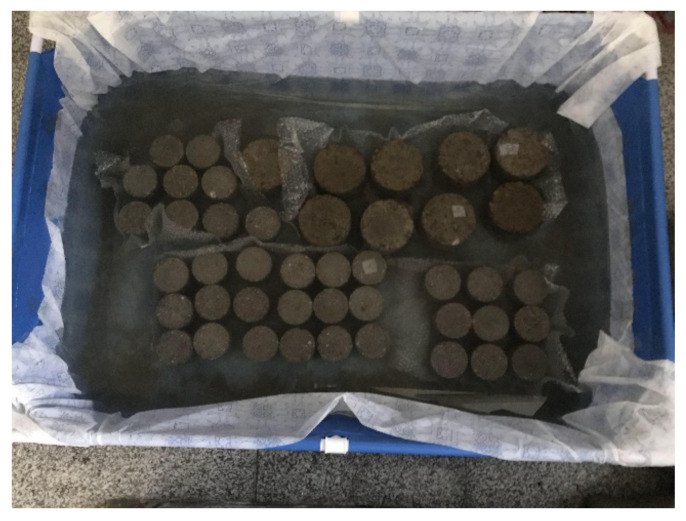
Sample saturation.

**Figure 5 materials-15-08614-f005:**
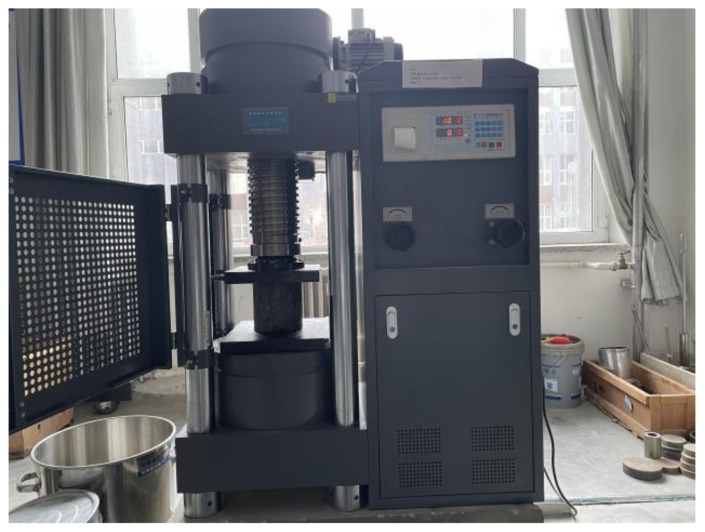
Fengyi RFP-03 Intelligent Dynamometer.

**Figure 6 materials-15-08614-f006:**
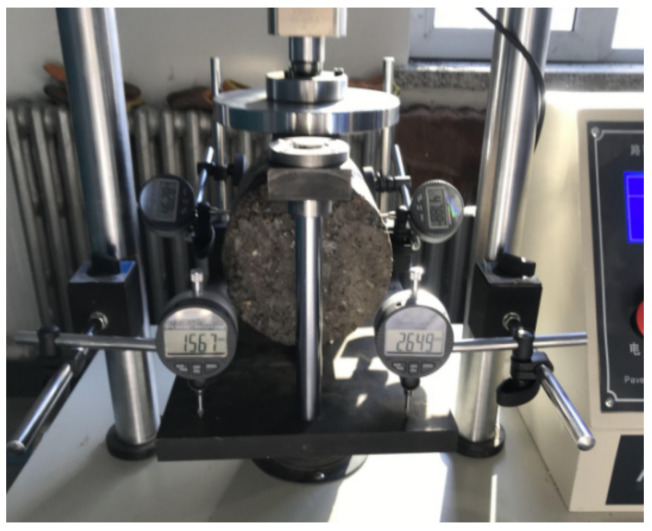
Lihuan pavement material strength tester LD-127.

**Figure 7 materials-15-08614-f007:**
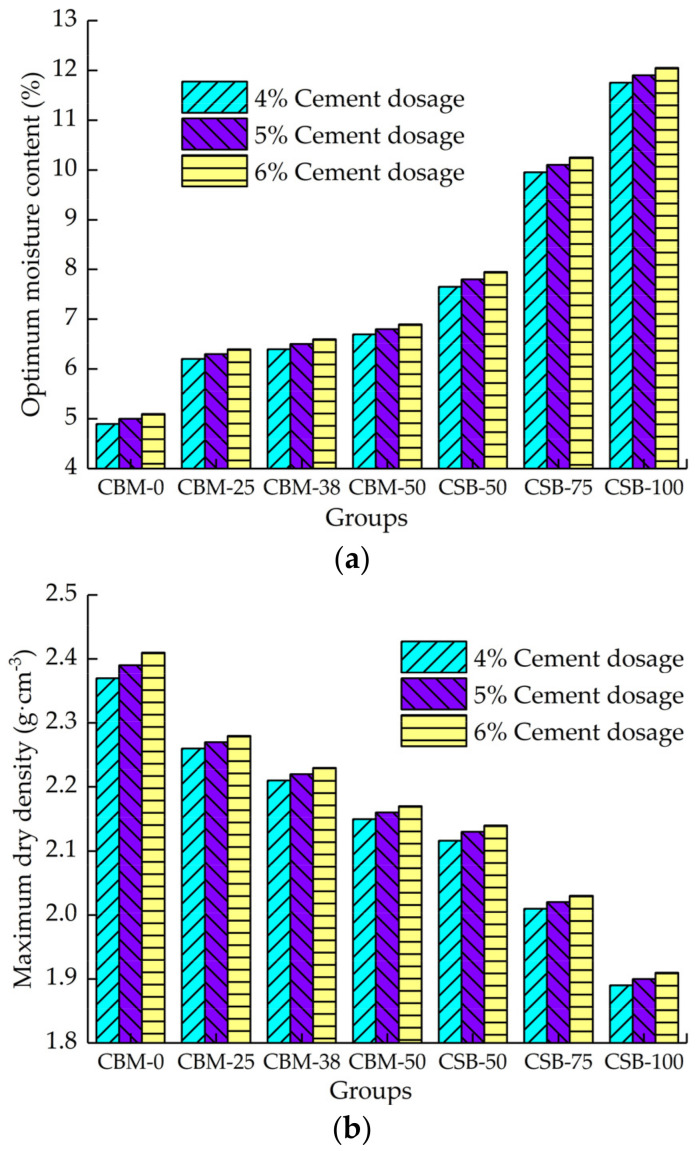
Test results of compaction test. (**a**) Optimum moisture content. (**b**) Maximum dry density.

**Figure 8 materials-15-08614-f008:**
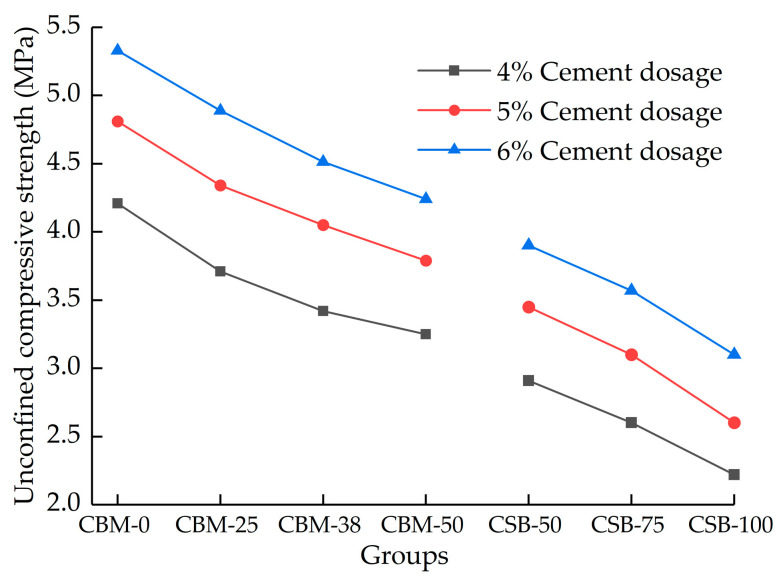
The unconfined compressive strength of mixture with different cement dosage at 7 days.

**Figure 9 materials-15-08614-f009:**
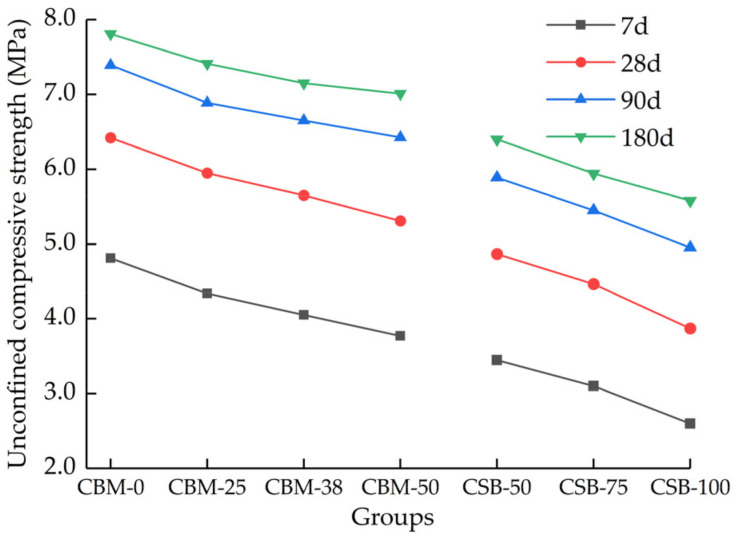
The unconfined compressive strength of mixture at different curing times.

**Figure 10 materials-15-08614-f010:**
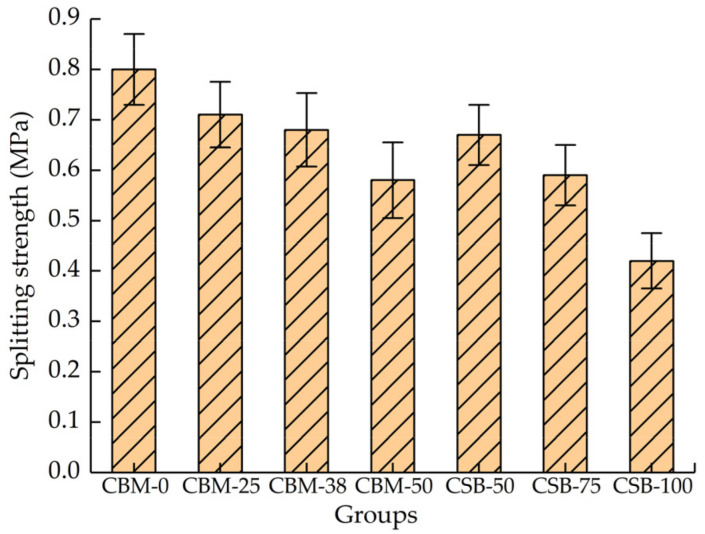
The splitting strength of mixture with 5% cement dosage after curing for 90 days.

**Figure 11 materials-15-08614-f011:**
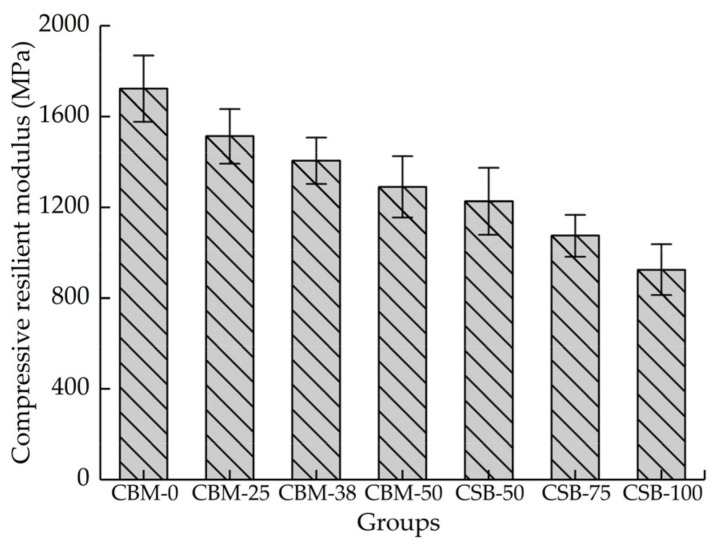
The compressive resilient modulus of mixture with 5% cement dosage after curing for 90 days.

**Figure 12 materials-15-08614-f012:**
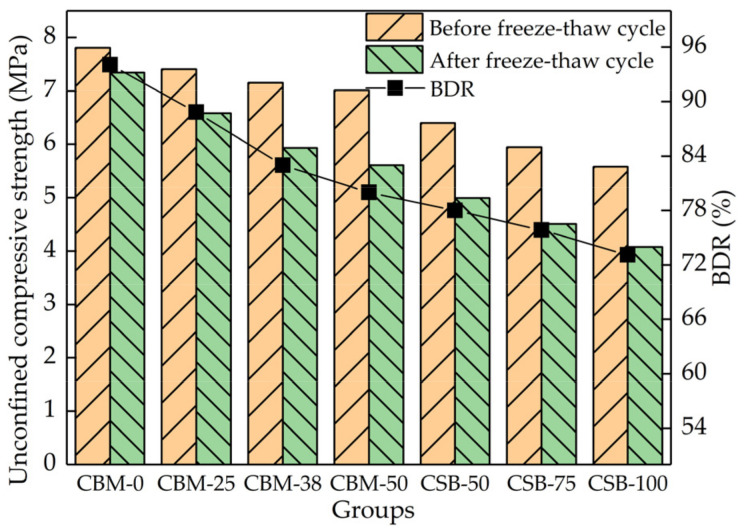
The unconfined compressive strength of mixture before and after F-T cycles.

**Figure 13 materials-15-08614-f013:**
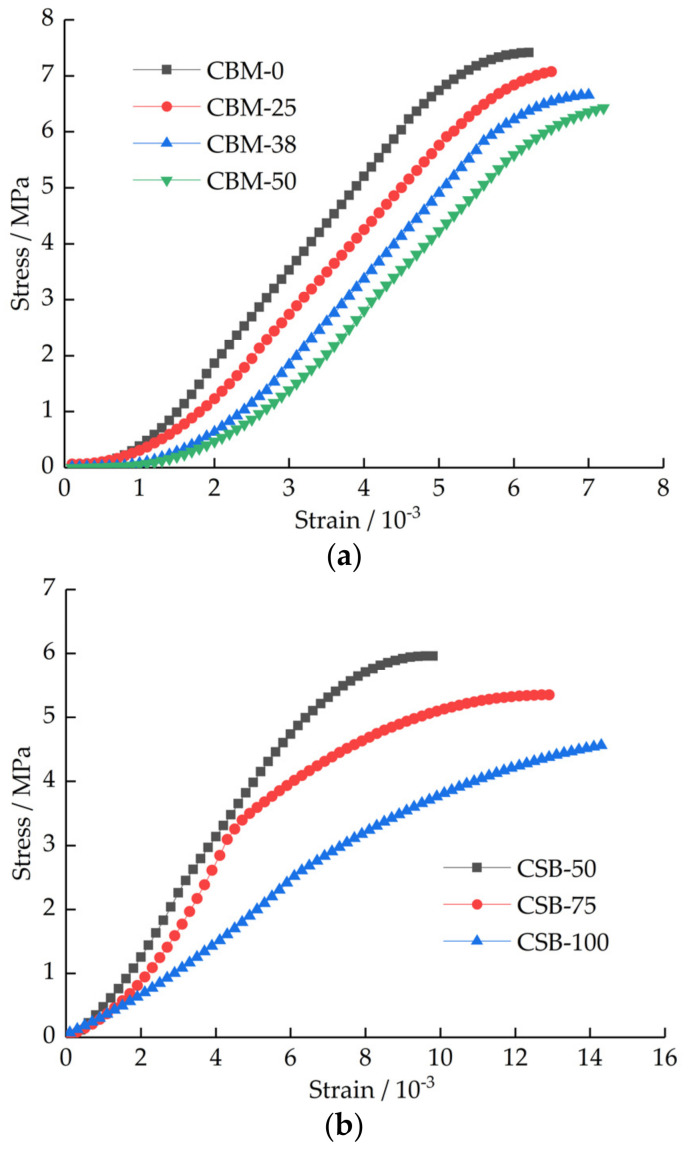
The stress–strain curve of mixture with 5% cement dosage after curing for 90 days. (**a**) CBM group. (**b**) CSB group.

**Figure 14 materials-15-08614-f014:**
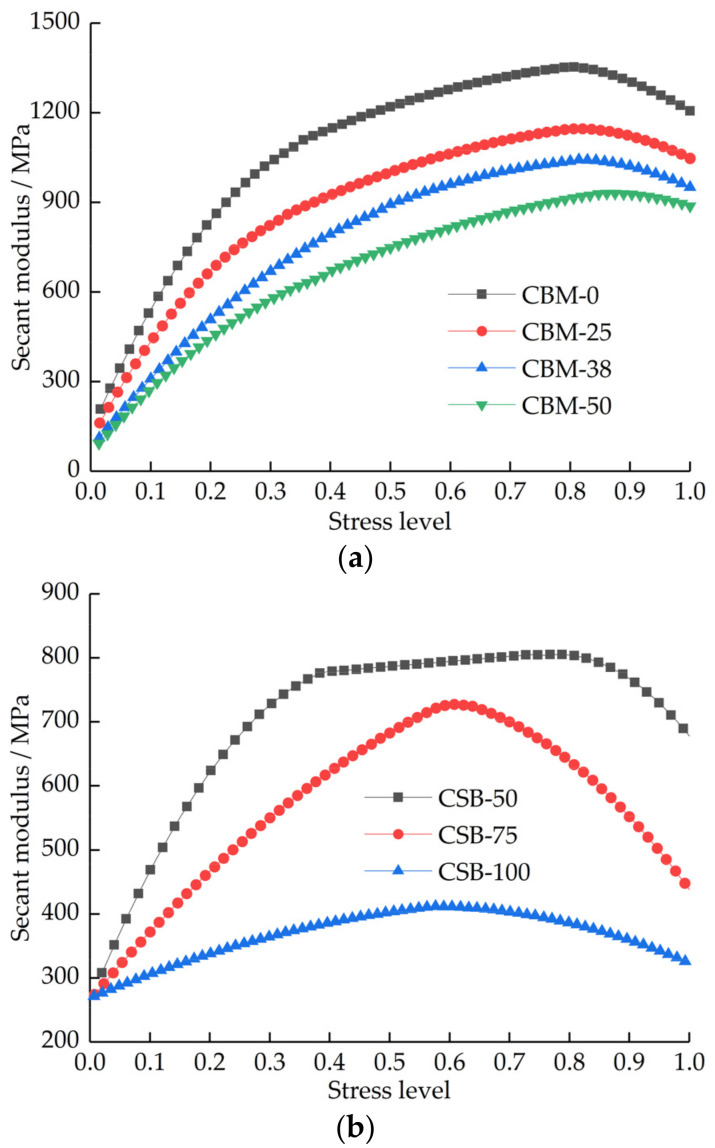
Curve of secant modulus under different stress levels. (**a**) CBM group. (**b**) CSB group.

**Figure 15 materials-15-08614-f015:**
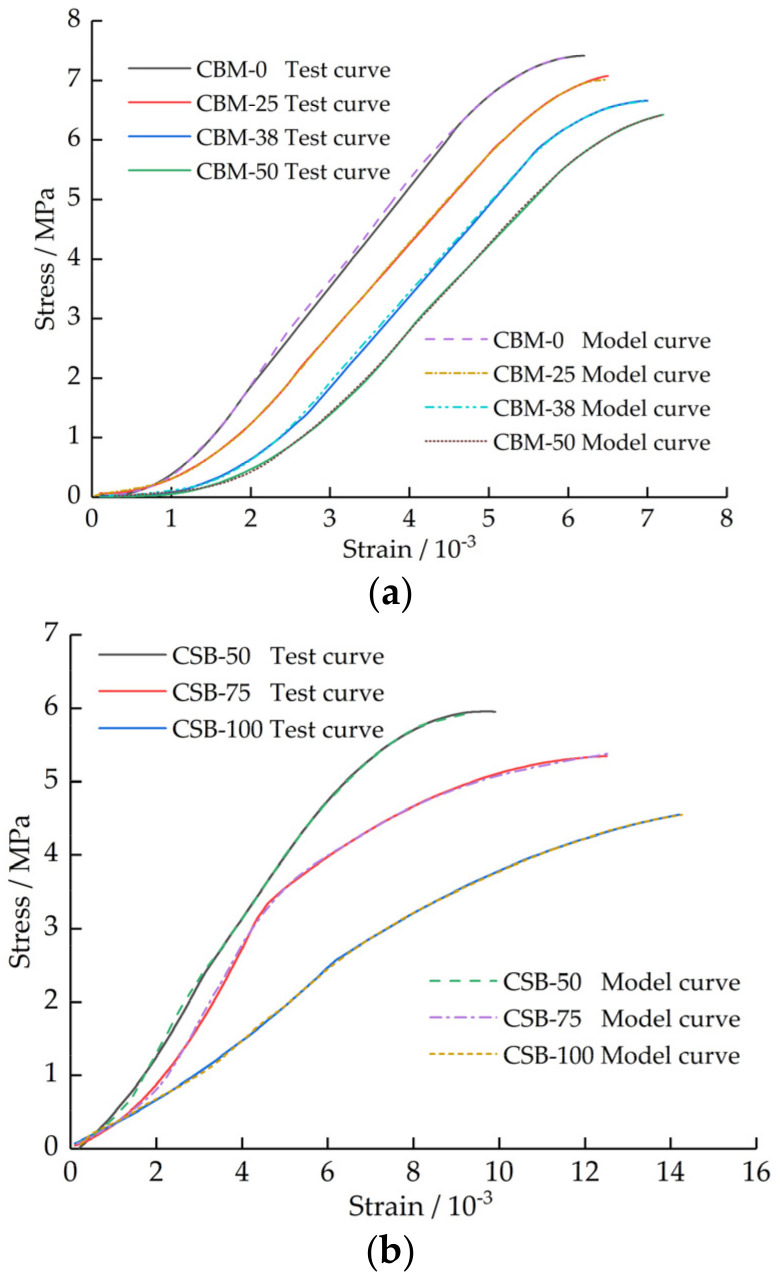
Comparison of test curve and model curve. (**a**) CBM group. (**b**) CSB group.

**Figure 16 materials-15-08614-f016:**
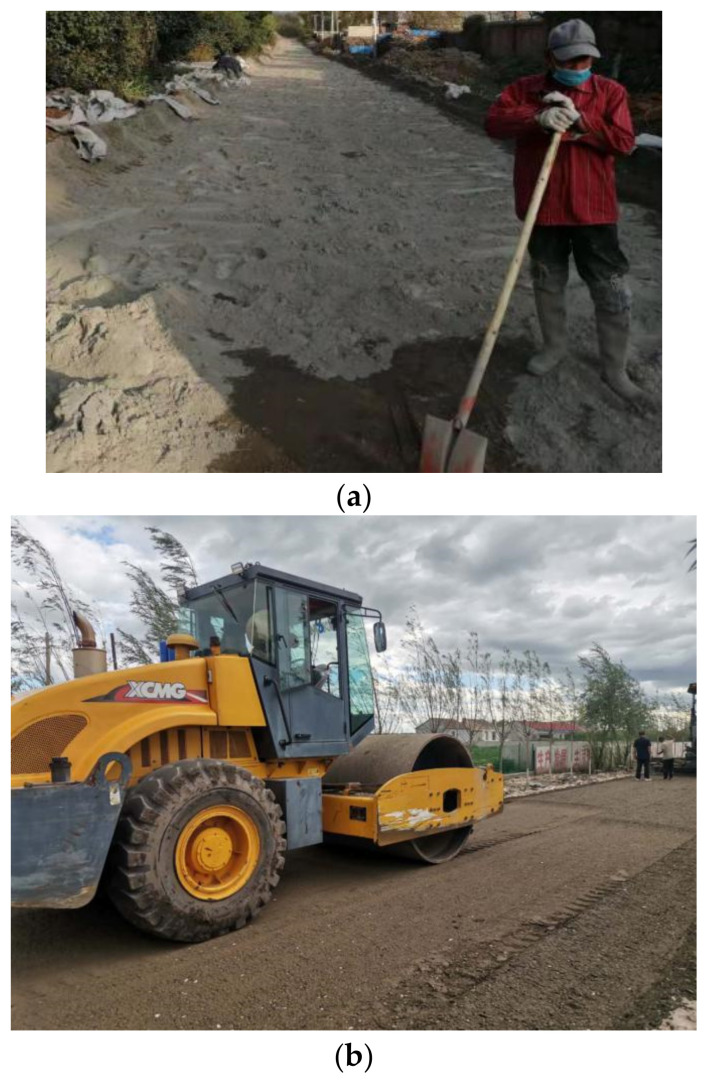
The main construction process. (**a**) Mixing. (**b**) Compaction.

**Figure 17 materials-15-08614-f017:**
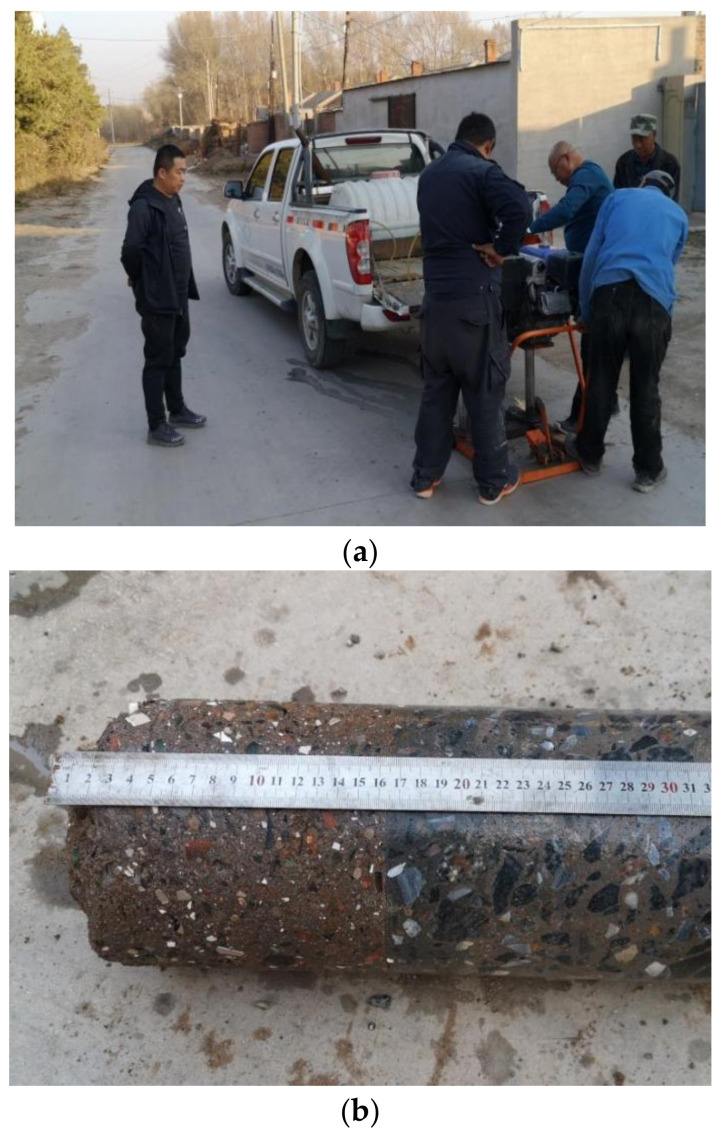
Drill core sampling. (**a**) Sampling. (**b**) Core sample.

**Table 1 materials-15-08614-t001:** Results of cement performance test.

Test Item		Experimentation Results	Specification
Fineness (%)		3.4	≤10
Stability (mm)		1.9	≤5
Setting time (min)	Initial setting time	225	≥180
	Final setting time	410	≥360, ≤600
Compressive strength (MPa)	3 days	13.2	≥10
	28 days	37.1	≥32.5
Flexural strength (MPa)	3 days	3.8	≥2.5
	28 days	7.2	≥5.5

**Table 2 materials-15-08614-t002:** Gradation of limestone aggregates.

Sieve Size (mm)	Percentage Passing (%)												
	31.5	26.5	19	16	13.2	9.5	4.75	2.36	1.18	0.6	0.3	0.15	0.075
1#	100	71.2	20.1	10.1	5.2	0.6	0.4	0.3	0.2	0.2	0.1	0.1	0.1
2#	100	100	100	59.0	33.4	9.5	2.3	1.6	1.5	1.3	1.2	0.9	0.4
3#	100	100	100	100	100	100	21.9	5.1	3.4	3.0	2.8	2.4	1.2
4#	100	100	100	100	100	100	100	85.1	77.8	69.2	39.8	7.2	2.0

**Table 3 materials-15-08614-t003:** Basic properties of limestone aggregates.

Sieve Size (mm)	Apparent Density (g/cm^3^)	Water Absorption (%)	Needle-like (%)	Crushing Value (%)	Organic Content (%)	Sulfate Content (%)
19~31.5	2.749	0.22	5.4	—	—	—
9.5~19	2.741	0.39	7.3	21.9	—	—
4.75~9.5	2.713	0.50	9.2	—	—	—
2.36~4.75	2.653	1.14	—	—	0.2	0.08
0~2.36	2.538	2.36	—	—		
Specification	—	—	≤18	≤35	<2	≤0.25

**Table 4 materials-15-08614-t004:** Basic properties of bottom ash.

Sieve Size (mm)	0~2.36	2.36~9.5	Specification
Apparent density (g/cm^3^)	2.546	2.511	—
Water absorption (%)	6.55	4.25	—
Liquid limit (%)	42.8		—
Plastic limit (%)	31.9		—
Plasticity index	10.9		≤17
Organic content (%)	0.90		<2
Sulfate content (%)	0.11		≤0.25

**Table 5 materials-15-08614-t005:** The chemical composition of MSWI BA.

Chemical Composition	SiO_2_	CaO	Al_2_O_3_	Fe_2_O_3_	Na_2_O	P_2_O_5_	SO_3_	K_2_O	MgO	CL	MnO	Others
Content (%)	39.44	31.61	9.61	7.53	2.33	2.30	2.26	1.68	1.72	0.84	0.17	0.52

**Table 6 materials-15-08614-t006:** Mix ratio design of mixtures.

Groups	Mixture Gradation				
	1#	2#	3#	4#	MSWI BA
CBM-0	25%	10%	30%	35%	0%
CBM-25	30%	10%	20%	15%	25%
CBM-38	25%	20%	17%	0%	38%
CBM-50	30%	15%	5%	0%	50%
CSB-50	-	-	-	50%	50%
CSB-75	-	-	-	25%	75%
CSB-100	-	-	-	0%	100%

**Table 7 materials-15-08614-t007:** Unconfined compressive strength standard of cement-stabilized material at 7 days (MPa).

Structural Layer	Highway Grade	Extremely Heavy and Special Heavy Traffic	Heavy Traffic	Medium and Light Traffic
Base	Expressways and first-grade highways	5.0~7.0	4.0~6.0	3.0~5.0
	Second- and below-order highways	4.0~6.0	3.0~5.0	2.0~4.0
Base course	Expressways and first-grade highways	3.0~5.0	2.5~4.5	2.0~4.0
	Second- and below-order highways	2.5~4.5	2.0~4.0	1.0~3.0

**Table 8 materials-15-08614-t008:** Related parameters of constitutive model.

**Groups**	** *a* **	** *b* **	** *c* **	** *k* **	** *d* **	** *F* ** ** _0_ **	** *h* **
CBM-0	584953	−256	0.0556	1668	−1.37	0.0200	9.98
CBM-25	340786	−100	0.0721	1511.1	−1.79	0.0078	14.10
CBM-38	309963	−369	0.1388	1530.6	−2.55	0.0087	10.61
CBM-50	250277	−333	0.1388	1384.5	−2.69	0.0082	22.04
CSB-50	111859	445	−0.082	857.9	−0.29	0.0128	4.62
CSB-75	124537	178	0.0246	-	-	0.0175	1.87
CSB-100	23541	262	0.0478	-	-	0.0200	4.23

## Data Availability

The data presented in this study are available on request from the corresponding author.

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
