# Peer review of "Research on Pavement Performance of Cement-Stabilized Municipal Solid Waste Incineration Bottom Ash Base"

_materials, 2022, doi:10.3390/ma15238614_

Round 1
Reviewer 1 Report
This article focuses on the Pavement performance of cement-stabilized municipal solid waste incineration bottom ash base. This manuscript is well organized. However, some minor issues need to be addressed before publication.
Abstract: Authors can condense the abstract and include major outcomes of the study
1. Recast line 11 and make it clearer.
2. Line 14 should read " 7 days unconfined …..”.
Line 29 should read “carbon peaking, and ……..”
3. Line 76, Please add a good justification of the work and indicate the novelty of the work. This is very important.
4. Line 81, Table, always indicates the day(s) instead of d
5. Line 93, Table 3, should read “apparent density (g/cm3)”.
6. Line 122, how can you define the plasticity behavior of bottom ash, which is having liquid limit 42.8 (intermediate compressibility)
7. Line 157, how can you define the slenderness ratio of the sample, and what about the maximum size of the particle in the blend?
8. Line 281-285, Need a scientific explanation. Based on the reviewer's knowledge, bottom ash is a lightweight material and has higher compressible nature. How can it develop dense phase and frictional resistance in the blend?
9. Line 291, should read “first-grade”
10. Line 314, Need better explanation. The compressive strength still increases greatly at the late curing stage with bottom ash.
11. Sections 3 and 4, the discussion of this work is needed to be improved. The authors need to rewrite the results and discussion section. Most results were only stated without comparing with other studies around the world. Authors need to compare the results of each parameter with similar studies around the world and also give reasons for differences between compared studies. The authors also need to reflect on the variations in the stress strains with the MSW bottom ash chemical composition.
12. Authors need to provide reasons for the increased levels of MSW bottom ash, even though there is a loss in the blend's performance.
13. Line-516, Avoid the usage of I or We in the whole manuscript
14. Section 6, more references will strengthen the discussion. Cite related studies and compare the sustainable utilization of MSW bottom ash in the pavements.
15. The conclusion should be rewritten and made briefer to show the major findings of the study and conclusions drawn.
16. Avoid references older than 5 yrs (>2017)
17. Finally, I suggest the authors thoroughly check the manuscript for spelling mistakes and correct it with a native speaker fluent in technical English to improve the language and its quality.
Reviewer 2 Report
- The title is in accord with article
- The manuscript adheres to the journal's standards after revision
- This article contains new aspects, but the authors must underline the major findings of their work and explain how this study represents a progress to other similar published papers. Please provide comparison with other articles
- The Abstract section refers to the study findings, methodologies, discussion as well as conclusion. The Abstract section must be improved. The Abstract should refer to the study findings, methodologies, discussion as well as conclusion. In this form the abstract is too generally
- The keywords permit found article in the current registers or indexes
- In the introduction it is not clearly described the state of the art of the investigated problem. More references are necessary. The references from last years are necessary for demonstrated that this study is actual
- The text can be understood by specialists from other domains
- The paper was written in standard, grammatically correct English, more corrections are necessary
- In Tables are presented necessary results
- The literature is sufficiently critical, current, and internationally evaluated
-The size of the article is appropriate to the content
- More discussions are necessary
Reviewer 3 Report
The present study investigates the influence of municipal solid waste incineration bottom ash content on the pavement performance. The paper is very interesting and presents valuable results in the domain of sustainable development in highway engineering. The study meets the Journal scope. In my opinion, the paper is of high scientific quality and very well organized. It can be accepted in the present form.
